# Tau Enhances Aggregation of S100A9 Protein and Further Association of Its Fibrils

**DOI:** 10.3390/ijms26188961

**Published:** 2025-09-15

**Authors:** Lukas Krasauskas, Dominykas Veiveris, Mantas Žiaunys, Darius Šulskis, Andrius Sakalauskas, Vytautas Smirnovas

**Affiliations:** Institute of Biotechnology, Life Sciences Center, Vilnius University, LT-10257 Vilnius, Lithuania; lukas.krasauskas@gmc.vu.lt (L.K.); dominykas.veiveris@gmc.vu.lt (D.V.); mantas.ziaunys@gmc.vu.lt (M.Ž.); darius.sulskis@gmc.vu.lt (D.Š.); vytautas.smirnovas@bti.vu.lt (V.S.)

**Keywords:** Tau, S100A9, amyloid, aggregation, cross-interaction

## Abstract

The formation and accumulation of amyloid fibrils is implicated as one of the main reasons for the onset and progression of several widespread neurodegenerative disorders, including Alzheimer’s and Parkinson’s diseases. Decades of effort to unravel the intricate mechanisms of amyloid aggregation have only led to limited success in developing potent treatment modalities. Generally, this failure is considered to be the result of our incomplete understanding of the processes governing protein transitions into these insoluble fibrillar structures. Recently, a growing number of studies have reported that multiple amyloidogenic proteins, including ones related to the most debilitating disorders, can cross-interact during aggregation. This process leads to different nucleation and fibril elongation rates, aggregate structures, and even their cytotoxicity. Despite this revelation, the entire amyloid interactome remains largely unexplored. In this work, we investigate the cross-interaction between the Alzheimer’s disease-related Tau protein and a pro-inflammatory S100A9 protein, which has recently been implicated as a possible modulator of amyloid aggregation. We show that Tau 2N4R enhances the amyloid aggregation propensity of S100A9 and mediates the self-association of the resulting fibrils, demonstrating this pairing’s potential role in the onset of neurodegenerative disorders.

## 1. Introduction

Protein aggregation into insoluble fibrillar aggregates has long been considered one of the main steps in the onset and progression of numerous degenerative diseases [1]. Among these, Alzheimer’s disease (AD) and Parkinson’s disease (PD) stand as the most widespread and debilitating afflictions [1,2]. Due to the ever-increasing average human lifespan, these age-related disorders are further rising in number and causing not only a major impact on the affected individuals but also straining worldwide healthcare systems [3,4,5]. Despite decades of intensive research into the mechanisms of protein amyloid aggregation, our current understanding of this process has only resulted in a handful of effective and approved treatment modalities [6,7,8]. In the case of most protein aggregation-related disorders, however, there is still a lack of preventative or disease-altering remedies.

The pathogenesis of AD involves the aggregation of amyloid-beta peptides and hyperphosphorylated Tau into extracellular plaques and intracellular tangles, respectively, while PD is characterized by the accumulation of α-synuclein into intracellular Lewy bodies [9,10]. During post-mortem studies on patients afflicted with amyloid-related disorders, a peculiar phenomenon was observed. The plaques formed by aggregated amyloid-beta peptide appeared to be inhomogeneous and contained various molecules, including other amyloidogenic proteins [11]. Further ex vivo and in vitro examinations revealed an even larger number of cross-interacting pairings, which include amyloid-beta with Tau [12,13] (related to AD), prion proteins [14] (prionopathies), and alpha-synuclein (PD) [15], as well as alpha-synuclein with Tau [16,17] and prion proteins [18]. Interestingly, even proteins that are not directly related to AD or PD, but still possess the capability of amyloid aggregation, have been shown to cross-interact with amyloid-beta, alpha-synuclein, and Tau [19]. Taken together, this data comprises only a part of the highly complex amyloidogenic proteins and peptide interactome, highlighting the need for further investigations into the matter.

Over the last decade, a new potential target has emerged in the field of neurodegenerative disorders—S100A9, which belongs to a family of calcium-binding pro-inflammatory S100 proteins [20] and is produced by neutrophils [21]. It has been observed that S100A9 can interact with the amyloid-beta peptide [22], as well as alpha-synuclein [23], suggesting its potential role in the onset of the two most prevalent neurodegenerative disorders. In both cases, S100A9 was able to co-aggregate and produce heterogenous amyloid clusters, as well as form transient electrostatic interactions with each protein while in its native conformation [22,23]. It was also shown that this protein was able to assemble into beta-sheet-rich worm-like fibers under near-physiological conditions [24], further adding to its probable role in amyloid diseases. Despite these reports, there is minimal information regarding its possible cross-interaction with another major player in AD—the microtubule-associated Tau protein. Based on the currently available literature, only S100B (a glial-specific member of the S100 family, which is found mostly in astrocytes [25]) has shown signs of interacting with Tau protein by inducing its hyperphosphorylation, and also acting as a chaperone to prevent Tau aggregation [26,27]. Furthermore, both relative amounts of S100B and S100A9 are increased in AD pathogenesis [28], but unlike S100B, S100A9 interactions with Tau are still unexplored.

Microtubule-associated Tau is an intrinsically disordered protein, which has six isoforms that are predominantly expressed in neurons [29,30]. In its non-aggregated state, Tau serves multiple physiological functions, which include regulation of microtubule dynamics, DNA stabilization, and participation in signaling pathways [31,32]. Based on the PAXdb protein abundance database [33], Tau colocalizes with S100A9 in multiple parts of the body with varying concentrations, including the cerebral cortex [34], frontal cortex [35], spinal cord [35], and lymph nodes [36]. In areas of inflammation, the concentration of S100A9 becomes significantly higher [37], resulting in an even higher probability of association events. The nature of their interaction is also a complex matter, as S100A9 can form amyloid-like aggregates [38], while Tau has been shown to bind with protein fibrillar structures both in vivo and in vitro [32,39,40].

In this work, we explore the potential interaction between S100A9 and the longest Tau 2N4R isoform under different conditions in vitro. We show that Tau can enhance the aggregation of S100A9 and act as an intermediary in the resulting fibril self-association without obtaining a beta-sheet structure itself. This peculiar, double effect of Tau on S100A9 adds another piece to the amyloid interactome puzzle and also reveals a possible mode of amyloid cluster formation.

## 2. Results

Based on our previous studies of amyloidogenic protein cross-interactions, three important aspects were observed in the case of Tau and S100A9. First, Tau interactions with other protein fibrils appeared to be governed mostly by electrostatic interactions [39]. Second, these interactions were conformation-dependent, where different fibril types could bind distinct amounts of Tau monomers [39]. Third, S100A9 had different effects on prion proteins, depending on whether it was in its native or aggregated state [41]. To account for all these factors, the following study was designed to incorporate a range of ionic strength conditions, as well as S100A9 aggregates, which were prepared using distinct buffer solutions.

For the initial cross-interaction examination, equimolar concentrations of non-aggregated recombinant (further referred to as “native conformation”) Tau and S100A9 were combined, and their aggregation was monitored under a range of ionic strength conditions, using thioflavin-T (ThT) fluorescence and optical density (OD) measurements (Figure 1A–C). ThT is an amyloid-specific fluorescent dye, which binds to the beta-sheet grooves of fibrils [42], while changes in sample turbidity or OD are characteristic of insoluble aggregate formation due to light scattering of larger particles [43]. As controls, solutions containing only one of each protein were tracked using the same experimental conditions. In the case of Tau control, the changes in fluorescence intensity were minimal, and only a relatively small increase was observed (Figure 1A). Interestingly, the same was not the case for the sample optical density. At the two lowest ionic strength conditions, the Tau sample OD_600_ experienced a significant (ANOVA Bonferroni means comparison, *p* < 0.01) multi-fold increase (Figure 1A). Such a high increase in OD was not detected in solutions containing 50 mM or 100 mM NaCl.

In the case of S100A9 control samples, all conditions resulted in a similar gradual increase of ThT fluorescence intensity (Figure 1B), with the 100 mM NaCl sample having a significantly lower end-point intensity (*p* < 0.05). Unlike Tau, none of the samples yielded any major changes in optical density, suggesting that if any structures formed, they were either too small to result in a detectable signal or their concentration was very low.

When both proteins were present in solution, there was a notable ThT signal increase from the very beginning of the reaction (Figure 1C). Under all conditions, the average end-point fluorescence intensity was higher than each of the controls, as well as the sum of both controls. The most peculiar result was obtained in the case of 0 mM NaCl, where the sample endpoint OD_600_ was multiple times higher than under any other investigated condition, despite no significant difference in its ThT intensity from the 20 mM NaCl concentration sample. Another notable factor was the OD_600_ of the 20 mM NaCl condition sample (Figure 1C), which was significantly lower than the control Tau under the same ionic strength (Figure 1A).

The disparity between the sample OD_600_ and its ThT fluorescence intensity could be explained by a number of possibilities. Tau is known for its ability to undergo liquid–liquid phase separation (LLPS) under quiescent conditions [44], which could result in the formation of condensates within the samples and subsequently increase their OD_600_. It may also form ThT-negative aggregates, which would have the same effect [45]. Cross-interactions with S100A9 may mitigate this process and result in a decreased OD_600_. To answer this question, the Tau control samples were examined using brightfield microscopy (Figure 1D). It was observed that the incubated Tau sample contained a small number of round particles (probable droplet-like structures), as well as similar low quantities of amorphous aggregates. This result indicates that condensation, as well as amorphous structure formation, may be the reason for the observed effect.

In the case of the disparity between OD_600_ and ThT intensity of the Tau and S100A9 samples under 0 mM NaCl conditions, one possible explanation would be a Tau-facilitated association of any formed S100A9 fibrils. This would markedly increase the sample’s light-scattering properties, without having a notable effect on the number of bound ThT molecules or their fluorescence intensity. Assembly of a fixed number of small particles into larger clusters would cause such an increase due to the non-linear dependence between particle volume and light-scattering intensity [46]. To investigate this possibility, the samples were scanned using transmission electron microscopy (TEM). The results showed that the control S100A9 fibril sample contained separate, worm-like fibrils, while S100A9 incubated with Tau displayed the presence of large aggregate clusters (Figure 1E). This observation supports the hypothesis that Tau facilitates S100A9 fibril self-association.

Since the presence of Tau resulted in a profound effect on S100A9 spontaneous aggregation at 0 mM NaCl, the process was further examined using higher concentrations of native conformation S100A9 under this ionic strength condition. During kinetic monitoring of the aggregation, it was observed that each of the selected S100A9 concentration control samples had lower ThT fluorescence emission intensities than their corresponding samples with Tau throughout the entire process (Figure 2A,B). Despite the Tau concentration being the same in all cases, there appeared to be an S100A9-concentration-dependent increase in the difference between the control and the cross-interaction samples (Figure 2B). At equimolar concentrations of both proteins, the difference in endpoint ThT intensity was approximately 20 a.u., while 37.5 µM S100A9 resulted in a 50 a.u. difference.

The sample optical density measurements presented an even more interesting outcome. All three concentration S100A9 samples displayed a minimal optical density both before and after the aggregation process (Figure 2C). However, the presence of Tau resulted in significant shifts of sample OD_600_, with 25 µM and 37.5 µM S100A9 samples surpassing the previously observed equimolar sample OD_600_ by two and three times, respectively. These results further support the idea that Tau plays a critical role in the amyloid formation process as well as the resulting aggregate clusterization.

To further investigate the propensity of Tau to induce S100A9 fibril self-association, the reaction kinetics under different ionic strength conditions were monitored using S100A9 fibrils generated under different conditions (using PBS or HEPES buffer solutions). The PBS variant fibril possessed a substantially larger tendency towards self-association and formed large aggregate clusters (Appendix A Figure A1A,B). In contrast, the HEPES variant fibril sample was composed of separate worm-like aggregates with a lower tendency toward cluster formation. The secondary structure of both fibril variants was quite similar (Appendix A Figure A1C,D), with the HEPES variant having a slightly higher parallel and anti-parallel beta-sheet content (1620 cm^−1^ and 1695 cm^−1^) and less random coil parts (1650 cm^−1^) [47]. To avoid any salt-induced effects or the presence of native/oligomeric conformation S100A9, both aggregate variants were centrifuged, washed, and resuspended in equal final concentrations using HEPES buffer solution.

When both fibril type control samples were incubated under different ionic strength conditions, a gradual decline in ThT fluorescence intensity was observed (Figure 3A,C). This is typically caused by ThT hydroxylation under neutral pH conditions [48]. Interestingly, while there were no notable changes in the HEPES variant samples, each PBS variant condition resulted in a decreased OD_600_, especially at low ionic strength conditions. This may be indicative of the PBS variant instability under sub-physiological ionic strength. Another notable aspect was the significantly higher ThT intensity, as well as the optical density of the PBS variant samples, when compared to the HEPES ones. Considering that the protein concentration in these samples was determined to be equal prior to the experimental procedure, it indicates that the PBS variant fibrils may have higher ThT-binding propensity, as well as self-association tendencies [39].

When the fibril samples were incubated in the presence of an equimolar Tau concentration, a peculiar disparity was observed between both variants. The HEPES fibrils initially had a low OD_600_ value, which increased upon incubation, especially in the case of 0 mM NaCl (Figure 3B). The same was true for the ThT intensity values, which were higher than the control samples (Figure 3A,B). In contrast, the PBS variant fibrils had a high initial OD_600_ value, which experienced the most significant decline upon incubation at low ionic strength conditions (Figure 3D). In this case, the ThT intensity values remained fairly similar to the control (Figure 3C,D). The highest tested ionic strength condition sample yielded almost no change in OD_600_, which was similar to the control. These results suggest that initially, Tau has a much more profound effect on the PBS variant S100A9 fibril self-association. However, over the incubation period, it slowly causes HEPES variant fibrils to self-associate, while PBS aggregates likely partially dissociate due to their instability under these low ionic strength conditions.

To gain deeper insight into the association tendencies, the 0 mM NaCl condition samples were imaged before (Figure 3E) and after (Figure 3F) incubation using TEM. The HEPES variant S100A9 fibrils were highly dispersed in both the control and the cross-interaction sample (Figure 3E). Upon incubation, the control sample aggregates experienced a low level of additional self-association, while in the case of the sample with Tau, incubation caused the formation of large fibril clusters (Figure 3F). In contrast, the PBS variant S100A9 fibrils already showed signs of cluster formation even in the control sample (Figure 3E). Upon the addition of Tau, as well as incubation, the large structures were still present (Figure 3E,F), and their quantity increased based on the previously observed changes in OD_600_. In this case, it is worth noting that TEM measurements only allow for a qualitative assessment of fibril morphologies and self-association tendencies, while providing limited information on their quantity or distribution.

Due to the TEM only imaging extremely small portions of the sample, where the fibrils are attached to grids, the Tau-induced self-association was further investigated via brightfield microscopy while in liquid phase. In the case of the control PBS variant fibril sample, the S100A9 aggregates were highly dispersed throughout each imaged area in small clusters (Figure 4A). When the sample contained Tau, there was a very apparent formation of larger aggregate assemblies, which ranged in size from several to tens of micrometers in diameter (Figure 4B). Their distribution throughout the sample was also fairly uneven. This complementary result further displays the ability of Tau to quickly cause the assembly of the PBS variant fibrils.

Since the effect of Tau on S100A9 fibril association appeared to be modulated by the solution’s ionic strength, it was further investigated whether the formed clusters could be dissociated with the addition of NaCl. Tau was combined with equimolar as well as two and 3 three times higher concentrations of PBS variant S100A9 fibrils. After a 30-min incubation at room temperature (22 °C), the samples were scanned and compared to the controls, which contained only S100A9 fibrils (Figure 5A–C). Interestingly, the OD_600_ difference between the samples and controls had a linear dependence on the concentration of S100A9 fibrils (Figure 5D). This suggested that the association-promoting properties of Tau extended beyond one-on-one interactions with S100A9 monomers.

Each of the prepared samples was titrated using a 1 M NaCl solution (50 mM HEPES, pH 7.4). The largest decrease in sample OD_600_ for all conditions was observed with the first addition of a small volume of the NaCl solution (Figure 5A–C). Each further addition resulted in a smaller decrease in OD_600_, until a plateau phase was reached at 8–10 µL added volume (~75–90 mM NaCl final concentration). In the case of the control samples, there appeared to be a small increase in the average OD_600_ upon the addition of the NaCl solution, despite an expected modest decrease due to the change in solution volume. This was likely the result of amyloid fibril self-association, promoted by the presence of salt ions, as we reported previously [49].

Curiously, the gap between each condition sample and the control OD_600_ did not disappear when the plateau was reached (Figure 5A–C and E). Additionally, the size of the difference followed a similar dependence on S100A9 fibril concentration as the 0 mM NaCl samples. These results suggest that, while higher ionic strength causes the majority of aggregate clusters to dissociate, some of them remain bound by non-electrostatic means.

Finally, an investigation was carried out to determine if there were any structural alterations to either native conformation Tau or S100A9 fibrils occurring during these association events. The aggregates formed during the reactions were pelleted and resuspended in heavy water (D_2_O), after which their FTIR spectra were acquired. Then, we attempted to reconstruct the sample spectra using only native conformation Tau and S100A9 fibril spectra. If this procedure resulted in a spectrum similar to the sample’s, it would indicate that the two components simply became bound to each other. Otherwise, if the spectra did not overlap, it would be a sign that the secondary structure of either Tau or S100A9 was altered during the reaction.

In the case of the HEPES and PBS variant S100A9 samples (Figure 6A,B), it was possible to almost perfectly reconstruct the sample spectra using only native conformation Tau and either S100A9 fibril variant spectra. This indicated that Tau did not undergo any major conformational changes and was likely incorporated into the aggregate clusters in its intrinsically disordered monomeric form. Interestingly, while there were no alterations in secondary structure, the pelleted HEPES variant fibril aggregates contained a much higher number of Tau molecules than the PBS variant. In the case of HEPES S100A9 fibrils, 8 S100A9 molecules were associated with one Tau, while in the case of the PBS variant, 45 S100A9 molecules were associated with one Tau. One possible explanation is that Tau has a higher affinity for the HEPES variant fibrils. Another likely explanation is that the PBS variant fibrils naturally form larger clusters, which then become associated via interactions with Tau and block further Tau binding.

The most interesting FTIR spectra were observed after S100A9 aggregation in the presence of an equimolar concentration of Tau (Figure 6C). In this case, the FTIR spectra displayed very low peaks associated with parallel beta-sheets (~1620 cm^−1^) and appeared to be mainly composed of a random-coil structure (1650 cm^−1^). The reconstruction of the spectra was also not entirely accurate, with the closest generated spectra having a notably lower peak associated with random coils (1650 cm^−1^), as well as a lower anti-parallel beta-sheet peak (1695 cm^−1^). In this case, it is possible that the cross-interaction with Tau resulted in a slightly different conformation of S100A9 fibrils or a higher number of aggregates with random coil secondary structures. The closest reconstruction also indicated that these assemblies contained only 2 S100A9 molecules for each Tau.

## 3. Discussion

The intrinsically disordered Tau protein’s primary function in vivo is the stabilization of microtubules [50]. It has also been reported to interact with a number of other proteins and nucleic acids, including the formation of DNA-stabilizing complexes [32]. In the case of neurodegenerative disorders, its aggregate formation is closely associated with the amyloid-beta peptide [31]. Amyloid structures formed by both proteins can accelerate each other’s aggregation and act synergistically to cause neurotoxic effects [51]. In vitro, Tau protein assembles into fibrillar structures with heparin [52] and binds with other amyloidogenic protein aggregates [17,39]. During this study, we have observed that Tau can also promote the amyloid formation process of S100A9, a proinflammatory protein that has recently been implicated as a possible contributor to the onset and progression of neurodegenerative disorders.

When analyzing the cross-interaction between non-aggregated Tau and S100A9, it was observed that the presence of Tau significantly enhances the process of S100A9 amyloid aggregation. This finding has a few possible explanations, including liquid-liquid phase separation (LLPS), as well as electrostatic interactions. The ability of Tau to assemble into high-concentration membrane-less liquid condensates is a well-known phenomenon, and signs of droplet-like structures were also observed in this study [53]. We have also recently reported that S100A9 can undergo LLPS and even form heterotypic droplets with another neurodegenerative disease-related protein—alpha-synuclein [54]. If the same event occurs for the Tau and S100A9 pairing, it can result in a significant increase in both protein local concentrations, which is a driving factor for S100A9 aggregation.

In the case of electrostatics, the cross-interaction between both proteins would be facilitated by their opposite charge at pH 7.4 (Tau pI—8.24, S100A9 pI—5.72, as calculated by the Expasy server [55]). This was clearly evident during experiments with non-aggregated Tau and S100A9 fibrils, where their co-presence resulted in an almost immediate formation of S100A9 aggregate clusters. Besides promoting the self-assembly of S100A9 fibrils, the bound positively charged Tau would partially negate the negative charge of S100A9 and facilitate easier surface-mediated secondary nucleation events. However, this electrostatic interaction appears to be highly sensitive to the environment’s ionic strength. The addition of even low amounts of NaCl to the solution (~50–100 mM) greatly reduced the formation of clusters by electrostatic shielding [56] and even caused the dissociation of preformed structures. This suggests that the process may be significantly impeded under physiological conditions. Neurons and glial cells can have ionic strengths ranging from 140 mM to 200 mM [57], while in the case of cerebrospinal fluid, it is typically around 150 mM [58].

The cluster formation tendencies of S100A9 in the presence of Tau may also be related to the structure of S100A9 fibrils. When they were generated in HEPES buffer solution, the aggregates were not prone to self-association. In contrast, when the fibrils were prepared in PBS and then resuspended in HEPES, they were noticeably more likely to form large aggregate clusters. This tendency persisted in the presence of Tau, where PBS-originated fibrils also resulted in significantly higher optical density samples. This phenomenon is likely related to the surface of the formed amyloid fibrils. Despite both types having a worm-like structure, small secondary structure differences were detected via FTIR spectroscopy. These minor distinctions may affect fibril association tendencies via surface charge or structure alterations, as was observed previously [39].

Another interesting observation made during this cross-interaction study was the lack of any significant structural alterations of Tau. When interacting with heparin or undergoing pathological aggregation, part of the intrinsically disordered protein obtains a beta-sheet structure [52]. Interactions with microtubules and DNA also result in a partial loss of Tau’s disordered structure [32]. However, when the FTIR spectra of Tau-S100A9 aggregates were analyzed, it revealed that Tau did not undergo any notable structural changes and remained disordered. These results suggest that Tau may act as an S100A9 fibril-binding agent through electrostatic interactions without forming any structure of its own. We have observed similar interactions of Tau with prion protein fibrils in a previous study [39]. Taking into consideration these results and reports of Tau within plaques of amyloid-beta aggregates, it is likely that this type of cross-interaction may be another function of Tau.

## 4. Materials and Methods

### 4.1. S100A9 Purification

Recombinant S100A9 was purified as described previously [41]. During the last stage of gel filtration, the protein was exchanged into either 50 mM HEPES buffer solution (pH 7.4) or PBS (pH 7.4), concentrated to 500 µM using Pierce protein concentrators (10 kDa cut-off, Thermo Scientific, Waltham, MA, USA), divided into 100 µL aliquots, and stored in test tubes at −80 °C. Prior to all further experimental procedures, the required volume of protein stock solution was thawed at room temperature (22 °C) before being immediately mixed with the reaction solutions. The purified S100A9 SDS-PAGE gel is shown in Figure A2.

### 4.2. Tau Purification

Plasmid encoding full-length Tau (2N4R) fused with N-terminal 6xHis-SUMO tag was transformed into One Shot BL21 Star (DE3) *Escherichia coli* (Thermo Scientific) cells by heat shock. The transformed cells were grown overnight in 100 mL of Lysogeny Broth (LB) medium containing kanamycin (50 μg/mL) at 37 °C. The culture was transferred to 100 mL of Terrific Broth (TB) medium with kanamycin (50 μg/mL) and grown at 37 °C until the optical density at 600 nm reached ∼0.8. Protein expression was induced by adding 1 mM IPTG and leaving the cells to grow at 37 °C for an additional 4 h.

Cells were collected by centrifugation and resuspended in lysis buffer (50 mM sodium phosphate, 1 M NaCl, pH 7.4) containing 1 mM phenylmethylsulfonyl fluoride (PMSF), one tablet of cOmplete Protease Inhibitor Cocktail (Roche, Basel, Switzerland) and lysozyme. Cells were lysed by sonication (Sonopuls, VS70T probe; Bandelin, Berlin, Germany) for 30 min at 40% amplitude (15 s on/15 s off cycle). The lysate was centrifuged (18,000 rpm, 45 min, 4 °C), and the supernatant was filtered through a 0.45 μm pore size filter.

Protein purification was conducted via immobilized metal ion affinity chromatography (IMAC) using a HisTrap HP 5ml column (Cytiva, Marlborough, MA, USA), pre-equilibrated with wash buffer (50 mM sodium phosphate, 1.0 M NaCl, pH 7.4). The cleared lysate was loaded into the column, and after the washing step, the protein was eluted using a stepwise gradient of 5% and 50% elution buffer (50 mM sodium phosphate, 1.0 M NaCl, 1M Imidazole, pH 7.4). Eluted fractions were checked by SDS-PAGE, and those containing 6xHis-SUMO-Tau fusion protein (59,116 kDa) were pooled for further purification.

The 6xHis-SUMO-Tau solution was dialyzed (8000 Da cut-off, Biodesign D106, Thermo Fisher, Waltham, MA, USA) against dialysis buffer (10 mM sodium phosphate buffer, pH 7.4) for 1 h. The catalytic domain of Sentrin-specific protease 1 (SENP1) was then added to cleave the 6xHis-SUMO tag, and the samples were dialyzed further overnight in freshly prepared dialysis buffer. The resulting protein solution was filtered through a 0.45 μm pore size filter and loaded onto a HiTrap Heparin HP (Cytiva) column. Protein was eluted by a linear gradient of 0–1 M NaCl in 10 mM sodium phosphate buffer (pH 7.4). Samples containing Tau protein were collected and subjected to reverse-IMAC purification, with the same protocol as the initial IMAC to remove the tag. All fractions were checked by SDS-PAGE for cleaved Tau (45,718 kDa) protein.

Prior to size exclusion chromatography (SEC), 10 mM EDTA and dithiothreitol (DTT) were added to the Tau protein. The protein was then concentrated (30 kDa cut-off, Merck, Darmstadt, Germany) and filtered through a 0.45 μm pore size filter. SEC was performed using a Tricorn 10/600 column (Cytiva) packed with Superdex 200 resin (Cytiva) and equilibrated with 50 mM HEPES (pH 7.4) buffer. The collected fractions were checked by SDS-PAGE, re-concentrated (30 kDa cut-off, Merck), and stored at −80 °C. The purified Tau SDS-PAGE gel is shown in Appendix A Figure A2.

### 4.3. S100A9 Fibril Preparation

Stock solutions of S100A9 were diluted to 200 µM using their respective buffer solutions (HEPES or PBS). The resulting reaction solutions were placed in a 96-well non-binding plate (cat. No 3881, Fisher Scientific, Waltham, MA, USA, 100 µL solution in each well). Nunc sealing tape was then placed over the plate, after which it was incubated at 37 °C for 48 h in a ClarioStar Plus plate reader (BMG Labtech, Ortenberg, Germany) under quiescent conditions. The incubation period was chosen based on previous studies, which examined S100A9 fibril formation under similar conditions [38,41].

After the aggregation reaction, the samples from the HEPES and PBS reaction solution wells were combined into two separate 2 mL non-binding test-tubes and centrifuged at 14,000× *g* for 30 min. The supernatants were carefully removed and replaced with a 50 mM HEPES buffer solution (pH 7.4) in both cases (to result in fibril samples with identical buffer solutions). The centrifugation step was repeated an additional time, and the aggregate pellets were resuspended in a three times lower volume of HEPES buffer solution. The concentration of the aggregates was determined by mixing an aliquot of each sample (10 µL) with a 7 M guanidinium hydrochloride solution (90 µL), incubating it for 30 min at 22 °C, and then scanning the sample absorbance at 280 nm using a Shimadzu UV-1800 (Kyoto, Japan) spectrophotometer (ε_280_ = 7090 M^−1^cm^−1^). Based on the obtained information, the aggregate solutions were diluted using 50 mM HEPES (pH 7.4) to result in 200 µM aggregate stock solutions (concentration was based on S100A9 monomers within the aggregate structure). The resulting solutions were stored at 4 °C prior to all further experimental procedures.

### 4.4. Cross-Interaction Monitoring

The stock solutions of non-aggregated Tau and S100A9, as well as S100A9 aggregates, were combined with 50 mM HEPES (pH 7.4), 50 mM HEPES with 1 M NaCl (pH 7.4), and 10 mM thioflavin-T (ThT) solutions to yield reaction mixtures containing 12.5 µM Tau and 12.5–37.5 µM of either S100A9 variant, 0–100 mM NaCl, and 50 µM ThT. For controls, the solutions contained only Tau or one of each S100A9 variant. The reaction solutions were then distributed to a 96-well non-binding plate (four repeats for every condition, 100 µL in each well). To account for the edge effect of plate incubation, the samples were placed in a non-sequential order (Appendix A Figure A3).

Prior to kinetic reaction monitoring, the optical density at 600 nm (OD_600_) of the samples was scanned using a ClarioStar Plus plate reader at 22 °C without the sealing tape. Afterwards, the plate was covered with Nunc sealing tape and incubated at 37 °C under quiescent conditions. ThT fluorescence measurements were taken every 10 min using 440 nm excitation and 480 nm emission wavelengths. After 96 h of incubation, the plate was cooled to 22 °C, the sealing tape was removed, and the sample OD_600_ was scanned as described previously.

### 4.5. Fourier-Transform Infrared Spectroscopy (FTIR)

After 96 h of incubation, the four repeats of each condition were recovered from the plate wells and combined. Part of each solution was taken for transmission electron microscopy measurements (50 µL total). The remaining solutions were centrifuged at 14,000× *g* for 30 min. The supernatants were carefully removed and used for an SDS-PAGE analysis. The aggregate pellets were resuspended in 200 µL of D_2_O and centrifuged at 14,000× *g* for 30 min. This resuspension and centrifugation procedure was repeated three times. After the final centrifugation step, the pellet was resuspended in 30 µL of D_2_O. The resulting samples were then scanned as described previously [59] using a Bruker Invenio S FTIR spectrometer (Bruker, MA, USA). To obtain non-aggregated Tau and S100A9 spectra, each protein stock solution was diluted using D_2_O (1:10 volume ratio) and concentrated using Pierce protein concentrators (10 kDa cut-off for S100A9 and 30 kDa for Tau). This dilution and concentration procedure was repeated three times. After the final concentration step, the sample FTIR spectra were scanned identically to the previous samples.

The obtained spectra were corrected by subtracting D_2_O and water vapor spectra, after which they were normalized to the same Amide I band between 1700 and 1595 cm^−1^. Their deconvolution, reconstruction, and comparisons were conducted as described previously [39] using GRAMS 8.0 software (Thermo Scientific).

### 4.6. Transmission Electron Microscopy (TEM)

An amount of 5 μL of each sample was applied to 300-mesh Formvar/carbon-supported copper grids (Agar Scientific, Rotherham, UK) and incubated for 1 min. Excess liquid was blotted with filter paper, and the grid was washed three times with 20 μL of distilled water, blotting between each wash. The same procedure was repeated with 5 μL of 2% (*w*/*v*) uranyl acetate for negative staining. TEM images were acquired using a Talos 120C (Thermo Fisher, MA, USA) microscope operating at 120 kV, equipped with a 4k × 4k Ceta CMOS camera (Thermo Fisher, MA, USA). Images were processed using Fiji software. Additional TEM images are available at Mendeley Data doi: 10.17632/k966dp6ngv.

### 4.7. Atomic Force Microscopy (AFM)

The HEPES and PBS variant fibrils were diluted to 12.5 µM using their respective buffer solutions. AFM sample preparations and measurements were performed as described previously [60]. In brief, the freshly cleaved mica was modified with (3-aminopropyl) triethoxysilane (APTES), after which 30 µL of each diluted sample was deposited on its surface. The mica was then washed with 2 mL of MilliQ H_2_O and dried under gentle airflow. AFM images were acquired using a Bruker Dimension Icon atomic force microscope. Data analysis was conducted using Gwyddion 2.57 software [61].

### 4.8. Brightfield Microscopy

Brightfield images of the samples were acquired as described previously [62]. In short, 15 μL aliquots of each sample were pipetted onto 1 mm thick glass slides (Fisher Scientific, cat. No. 11572203), covered with 0.18 mm coverslips (Fisher Scientific, cat. No. 17244914), and imaged using an Olympus IX83 microscope (Evident Corporation, Tokyo Japan) with a 40× objective (Olympus LUCPLANFL N 40× Long Working Distance Objective). The resulting images were processed and analyzed using Fiji software [63]. Additional brightfield images are available at Mendeley Data doi: 10.17632/k966dp6ngv.

### 4.9. Statistics

All data were analyzed using Origin Software (OriginLab, Northampton, MA, USA) ANOVA Bonferroni means comparison. Statistically significant differences described in this work correspond to * *p* < 0.05, ** *p* < 0.01, *** *p* < 0.001 from three or four technical repeats.

## 5. Conclusions

Tau enhances the aggregation of S100A9 and acts as an intermediary in the resulting fibril self-association. However, Tau itself does not undergo any notable structural alterations, with electrostatic interactions being the main force in the assembly of these heterogenous aggregate clusters.

## Figures and Tables

**Figure 1 ijms-26-08961-f001:**
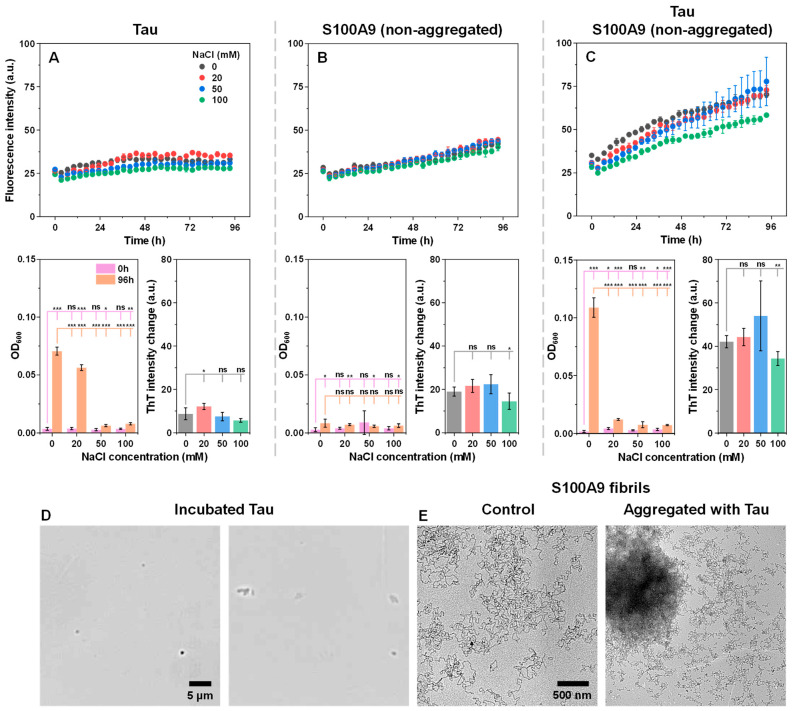
Native conformation Tau and S100A9 cross-interaction monitoring. Thioflavin-T (ThT) fluorescence intensity changes over time for Tau control (**A**), S100A9 control (**B**), and Tau with S100A9 (**C**). Kinetics were monitored using equimolar protein concentrations (12.5 µM), 50 µM ThT, under constant 37 °C incubation and quiescent conditions in 50 mM HEPES buffer solution. Each data point is the average of four technical repeats; error bars are for one standard deviation; statistical analysis was conducted using ANOVA Bonferroni means comparison (ns—not significant, * *p* < 0.05, ** *p* < 0.01, *** *p* < 0.001). Panels below their corresponding kinetic data show sample OD_600_ values at the start and end of the reaction (four technical repeats; error bars are for one standard deviation). OD_600_ measurements were performed at 22 °C in both cases. Sample ThT fluorescence intensity changes over the duration of the procedure display the intensity difference between the endpoint and 10-min measurement (when the sample reached thermal equilibrium, four technical repeats; error bars are for one standard deviation). Brightfield microscopy images of the incubated 12.5 µM Tau sample under 0 mM NaCl conditions (**D**); the scale bar is 5 μm. Transmission electron microscopy images of 12.5 µM S100A9 fibrils formed in the absence or presence of 12.5 µM Tau under 0 mM NaCl conditions (**E**); the scale bar is 500 nm for both images.

**Figure 2 ijms-26-08961-f002:**
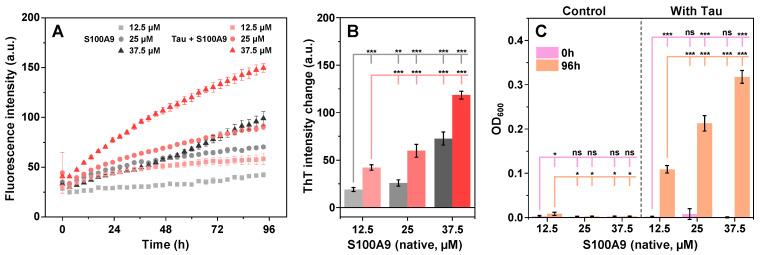
Kinetics of native conformation S100A9 aggregation in the absence or presence of Tau. Different concentration (12.5 µM to 37.5 µM) native conformation S100A9 sample aggregation ThT fluorescence kinetics in the absence (grey points) or presence (red points) of Tau (**A**). Tau concentration was 12.5 µM in each of the different concentration S100A9 samples. Each data point is the average of four technical repeats, and error bars are for one standard deviation. Panel (**B)** displays sample ThT fluorescence intensity changes over the duration of the procedure shown in panel A (color-coded to kinetic data, four technical repeats; error bars are for one standard deviation). The intensity difference is shown between the endpoint and the 10-min measurement (when the sample reached thermal equilibrium). Panel (**C**) displays sample OD_600_ values at the start and end of the reaction shown in panel A (four technical repeats; error bars are for one standard deviation; statistical analysis was conducted using ANOVA Bonferroni means comparison (ns—not significant, * *p* < 0.05, ** *p* < 0.01, *** *p* < 0.001). OD_600_ measurements were performed at 22 °C in both cases.

**Figure 3 ijms-26-08961-f003:**
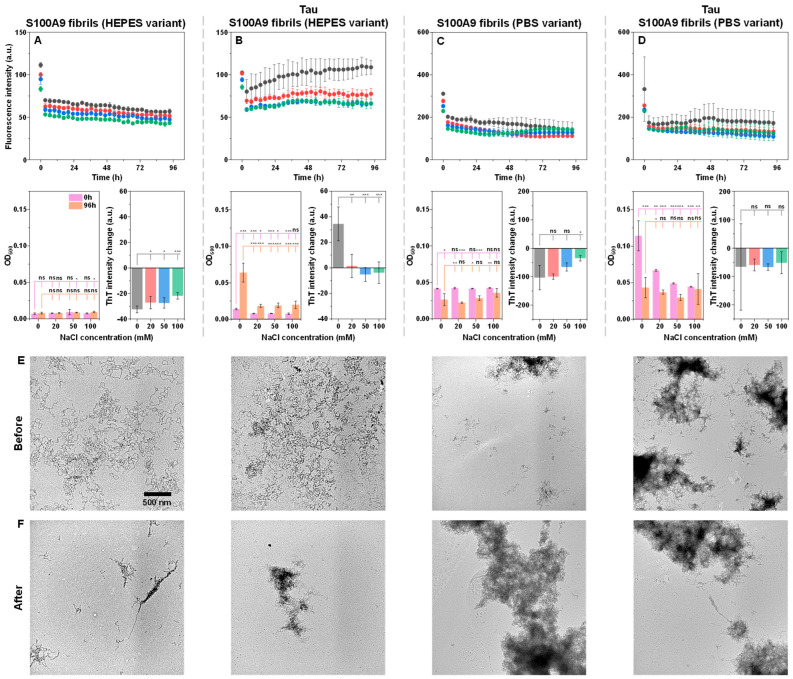
Native conformation Tau and S100A9 fibril variant cross-interaction monitoring. Thioflavin-T (ThT) fluorescence intensity changes over time for HEPES variant S100A9 fibrils in the absence (**A**) or presence (**B**) of Tau and PBS variant S100A9 fibrils in the absence (**C**) or presence (**D**) of Tau (each color represents technical repeat). Kinetics were monitored using equimolar protein concentrations (12.5 µM, based on aggregated monomer concentration), 50 µM ThT, under constant 37 °C incubation and quiescent conditions. Each data point is the average of four technical repeats; error bars are for one standard deviation. Panels below their corresponding kinetic data show sample OD_600_ values at the start and end of the reaction (four technical repeats; error bars are for one standard deviation; statistical analysis was conducted using ANOVA Bonferroni means comparison (ns—not significant, * *p* < 0.05, ** *p* < 0.01, *** *p* < 0.001). OD_600_ measurements were performed at 22 °C in both cases. Sample ThT fluorescence intensity changes over the duration of the procedure display the intensity difference between the endpoint and 10-min measurement (when the sample reached thermal equilibrium, four technical repeats; error bars are for one standard deviation). Transmission electron microscopy images of samples before (**E**) and after (**F**) incubation. Each condition’s sample images are shown directly below their corresponding column. The scale bar is 500 nm for all images.

**Figure 4 ijms-26-08961-f004:**
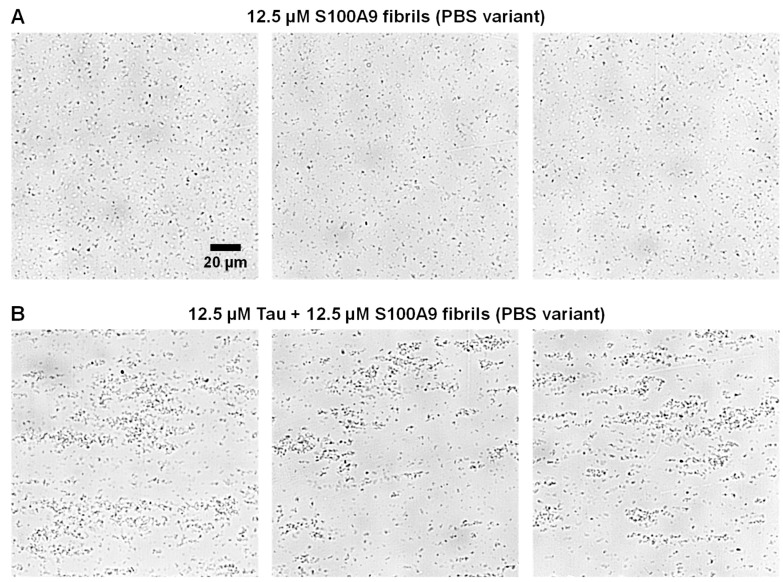
Brightfield microscopy images of 12.5 µM S100A9 fibril samples in the absence (**A**) or presence (**B**) of 12.5 µM Tau in 50 mM HEPES buffer solution without NaCl. Images were acquired at different locations of each sample after 30 min of incubation at room temperature (22 °C) under quiescent conditions. The scale bar is 20 μm for all images.

**Figure 5 ijms-26-08961-f005:**
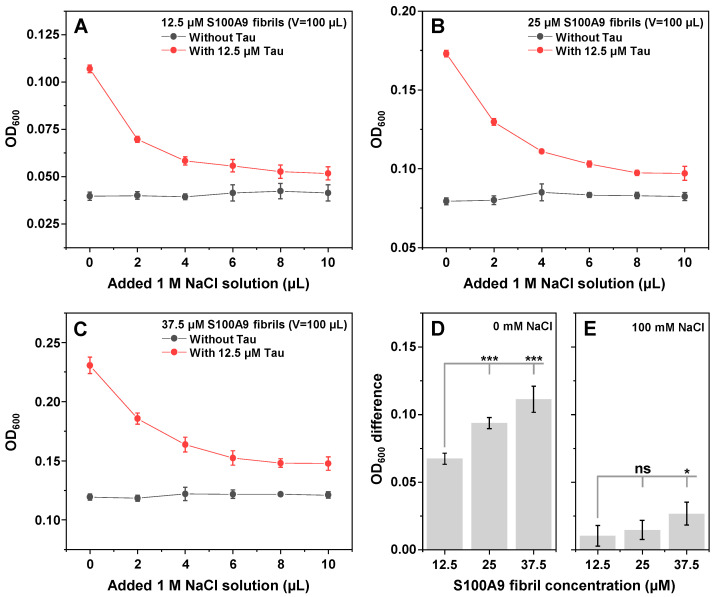
Tau–S100A9 fibril cluster dissociation. Optical density dependence on ionic strength of Tau and S100A9 fibril samples ((**A**)—12.5 µM, (**B**)—25 µM, (**C**)—37.5 µM S100A9 fibrils; each sample contained 12.5 µM Tau). OD_600_ differences between Tau–S100A9 and S100A9 samples at 0 mM NaCl (**D**) and 100 mM NaCl (**E**). All measurements were performed in a 50 mM HEPES buffer solution; titration was conducted using a 50 mM HEPES buffer solution containing 1 M NaCl. Each data point is the average of three technical repeats; error bars are for one standard deviation; statistical analysis was conducted using ANOVA Bonferroni means comparison (ns—not significant, * *p* < 0.05, *** *p* < 0.001.

**Figure 6 ijms-26-08961-f006:**
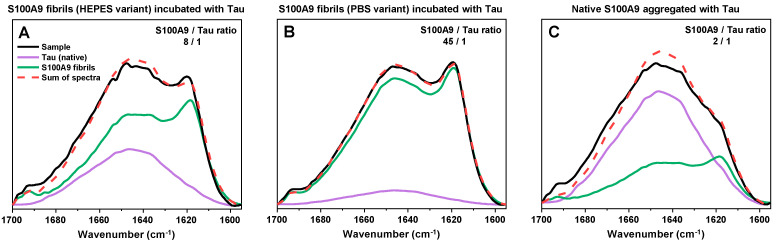
S100A9–Tau sample FTIR spectra and their reconstruction using native conformation Tau and S100A9 fibril spectra. FTIR spectra of Tau aggregates formed with HEPES variant (**A**), PBS variant (**B**), and native (**C**) S100A9. Reconstruction of sample spectra (red dashed line) was conducted as described in Section 4 using the spectra of native conformation Tau (purple) and either variant S100A9 fibril spectra (green). The S100A9 and Tau ratio (shown in each panel’s top right corner) was calculated based on the number of amide bonds within each protein and their respective spectra volume in the reconstruction.

## Data Availability

All data are available at Mendeley Data doi: 10.17632/k966dp6ngv.1.

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
