# Peer review of "Tau Enhances Aggregation of S100A9 Protein and Further Association of Its Fibrils"

_ijms, 2025, doi:10.3390/ijms26188961_

Round 1
Reviewer 1 Report
Comments and Suggestions for Authors
This manuscript describes potential interactions between S100A9 and tau under various in vitro conditions. Below are some comments that will help improve the manuscript.
Line 28-29. Fibrillar aggregate formation is not characteristic of multiple sclerosis. The pathogenesis of multiple sclerosis is associated with the formation of antibodies to myelin sheath proteins.
Line 41. Please add details on the pathogenesis of Alzheimer's disease and Parkinson's disease. For a review of the pathogenesis of Alzheimer's disease, please see the article: 10.3390/brainsci15050486 .
Line 53. Please add details on the interaction of S100A9 with beta-amyloid peptide and with alpha-synuclein. How does this interaction occur? What is the result of this interaction?
Lane 60. Please add characterization of the S100B protein.
Line 92. Please specify what is the aggregation tracking method based on?
Line 313. Please specify the relationship of tau aggregates to beta-amyloid peptide [26].
Line 322-323. Please describe in more detail the process of liquid-liquid phase separation, as well as electrostatic interactions.
Line 363. Aggregation studies in different salt buffers differ significantly from physiological conditions. How can these data be related to in vivo data? Please add information about physiological conditions and aggregation in the brain.
Line 495. The term “pro-inflammatory” should not be used here
Author Response
Author response in green.
We would like to thank both Reviewers for their time and effort in reviewing our manuscript and for offering suggestions on how to improve it. We have included detailed answers to all the queries and comments in this document.
Reviewer 1
This manuscript describes potential interactions between S100A9 and tau under various in vitro conditions. Below are some comments that will help improve the manuscript.
Line 28-29. Fibrillar aggregate formation is not characteristic of multiple sclerosis. The pathogenesis of multiple sclerosis is associated with the formation of antibodies to myelin sheath proteins.
There are reports of a complex correlation between amyloid aggregates and multiple sclerosis. However, we agree with the reviewer that this statement may create a slight confusion. We have modified this statement to exclude multiple sclerosis.
Line 41. Please add details on the pathogenesis of Alzheimer's disease and Parkinson's disease. For a review of the pathogenesis of Alzheimer's disease, please see the article: 10.3390/brainsci15050486 .
We have added details regarding AD and PD pathogenesis, as well as cited the aforementioned publication.
Line 53. Please add details on the interaction of S100A9 with beta-amyloid peptide and with alpha-synuclein. How does this interaction occur? What is the result of this interaction?
We have added details that S100A9 can co-aggregate, as well as form transient electrostatic interactions with both amyloid-beta, as well as alpha-synuclein while in its native state.
Lane 60. Please add characterization of the S100B protein.
We have added information about the specifics of S100B protein in the Introduction.
Line 92. Please specify what is the aggregation tracking method based on?
We have added an explanation and references for the two used methods in this part of the text. ThT is an amyloid-specific fluorescent dye, which binds to the beta-sheet grooves of fibrils, while changes in sample OD are characteristic of insoluble aggregate formation due to light scattering of larger particles
Line 313. Please specify the relationship of tau aggregates to beta-amyloid peptide [26].
We have specified the relationship of Tau aggregates to beta-amyloid peptide in the Introduction.
Line 322-323. Please describe in more detail the process of liquid-liquid phase separation, as well as electrostatic interactions.
Thank you for the suggestion, however, we believe that LLPS and electrostatic interactions are fairly self-explanatory in the field of protein cross-interactions. The manuscript contains citations to these specific effects “The ability of Tau to assemble into high concentration membraneless liquid condensates is a well-known phenomenon and signs of droplet-like structures were also observed in this study [48].” And “In the case of electrostatics, the cross-interaction between both proteins would be facilitated by their opposite charge at pH 7.4 (Tau pI – 8.24, S100A9 pI – 5.72, as calculated by the Expasy server [50]).”
Line 363. Aggregation studies in different salt buffers differ significantly from physiological conditions. How can these data be related to in vivo data? Please add information about physiological conditions and aggregation in the brain.
We have added information that neurons and glial cells have an ionic strength ranging from 140 mM to 200 mM, while cerebrospinal fluid has an ionic strength of 150 mM. We also have a statement in the manuscript, that “The addition of even low amounts of NaCl to the solution (~50 – 100 mM) greatly reduced the formation of clusters and even caused the dissociation of preformed structures. This suggests that the process may be significantly impeded under physiological conditions.”
Line 495. The term “pro-inflammatory” should not be used here
The term was removed from this part of the manuscript.
Reviewer 2 Report
Comments and Suggestions for Authors
In this work, Authors investigated the cross-interaction between the Alzheimer’s disease-related Tau protein and a pro-inflammatory S100A9 protein. They showed that Tau 2N4R enhances the amyloid aggregation propensity of S100A9 and mediates the self-association of the resulting fibrils. However, the results in the manuscript cannot fully confirm this conclusion. Besides, the presentation and writing of the manuscript result are poor. Therefore, I do not recommend it be published in International Journal of Molecular Sciences in its current form. My comments are following:
- In addition to fluorescence and electron microscopy results, the authors should use western blot to detect the state of Tau Enhances Aggregation of S100A9 Protein.
- Why author using the optical density at 600 nm (OD600) to evaluate the status of the aggregation, as it is a commonly used method for determining bacterial concentration.
- The“significantly”and “no significantly”appeared many times in the manuscript, but no statistical analysis was conducted in all the Figure results. I have no idea how the author reached this conclusion.
- How does the author purify native tau and S100A9 protein? In the manuscript, the author described the purification method using prokaryotic expression.
- No significant aggregation of S100A9 is observed in images of TEM in the Figure 1E, I am very confused about how the author got Tau facilitates S100A9 fibril self-association.
- What are the concentrations of tau protein in Figure 2B and C.
- The results in Figures 3A-D indicate that tau is not a necessary condition for enhancing the aggregation of S100A9, as the ThT intensity of S100A9 does not change significantly even in the presence of tau (as shown in Figure 3D)
- Is it possible that the Tau Enhances Aggregation of S100A9 Protein claimed by the author is a combination of Tau and S100A9
- Why higher ionic strength causes the majority of aggregate clusters to dissociate? There is such a high higher ionic strength under physiological conditions?If not, what is the significance of the research?
- The result section fails to present the results in the picture very well
- Figure legend cannot describe and summarize pictures well
Author Response
Author response in green.
We would like to thank both Reviewers for their time and effort in reviewing our manuscript and for offering suggestions on how to improve it. We have included detailed answers to all the queries and comments in this document.
Reviewer 2
In this work, Authors investigated the cross-interaction between the Alzheimer’s disease-related Tau protein and a pro-inflammatory S100A9 protein. They showed that Tau 2N4R enhances the amyloid aggregation propensity of S100A9 and mediates the self-association of the resulting fibrils. However, the results in the manuscript cannot fully confirm this conclusion. Besides, the presentation and writing of the manuscript result are poor. Therefore, I do not recommend it be published in International Journal of Molecular Sciences in its current form. My comments are following:
1. In addition to fluorescence and electron microscopy results, the authors should use western blot to detect the state of Tau Enhances Aggregation of S100A9 Protein.
Thank you for the suggestion, however, we already use electron microscopy, fluorescence and optical density measurements, as well as FTIR spectroscopy, which all point towards the same conclusion. Western blot of the samples would not provide any additional relevant information, since the main observed aspect is the formation of large heterogenous aggregate clusters, rather than various forms of small oligomeric structures.
2. Why author using the optical density at 600 nm (OD600) to evaluate the status of the aggregation, as it is a commonly used method for determining bacterial concentration.
Optical density measurements are also a commonly used method to detect protein aggregation or fibril self-association into larger clusters. We have added additional emphasis and explanation on this matter in the Result section. Since we used freshly purified proteins, which did not contain any procaryotic cells, the OD change reflects the association or dissociation of smaller or larger protein clusters
3. The“significantly”and “no significantly”appeared many times in the manuscript, but no statistical analysis was conducted in all the Figure results. I have no idea how the author reached this conclusion.
Statistical analysis was done using ANOVA Bonferroni means comparison, significant differences were considered when p<0.01. Due to the large number of data points, the significance is mentioned only for relevant points, rather than adding many significance markers within each graph. We have added a statement regarding this at the first mention of significant differences in the Result section. The Materials and Methods section also has a paragraph describing the software and procedure.
We have also included statements regarding the number of repeats and error bars in each Figure Caption.
4. How does the author purify native tau and S100A9 protein? In the manuscript, the author described the purification method using prokaryotic expression.
The recombinant versions of both proteins are produced in E.Coli cells and purified as described in the Materials and Methods section. The use of the term “native” and “native conformation” in the manuscript refers to the conformational state of the protein (non-aggregated). We have modified the manuscript to emphasize that the conformation is native, not that the protein is extracted from eukaryotic cells.
5. No significant aggregation of S100A9 is observed in images of TEM in the Figure 1E, I am very confused about how the author got Tau facilitates S100A9 fibril self-association.
S100A9 forms worm-like aggregates with a cross-section that is lower than most other protein fibrils. The control image with S100A9 fibrils in Figure 1E shows aggregates that are usually produced during S100A9 amyloid aggregation. The “Aggregated with Tau” image shows large aggregate cluster formation from the worm-like fibril interaction with Tau. This is further supported by changes in sample optical density, where it increases despite the concentration of the protein remaining the same, which indicates large particle formation.
6. What are the concentrations of tau protein in Figure 2B and C.
The Figure caption was updated with this information. The Tau concentration is 12.5 uM for all samples which contain it.
7. The results in Figures 3A-D indicate that tau is not a necessary condition for enhancing the aggregation of S100A9, as the ThT intensity of S100A9 does not change significantly even in the presence of tau (as shown in Figure 3D)
Figure 3 represents data of Tau interactions with S100A9 fibrils (pre-aggregated, centrifuged and resuspended into HEPES buffer solution). Since the fibrils are already formed and there should be minimal amounts of residual S100A9 monomers, no increase in ThT intensity is expected. This measurement is done to show that Tau does not form fibrils by itself under these conditions and that S100A9 fibrils only self-associate (without an increase in their amount).
8. Is it possible that the Tau Enhances Aggregation of S100A9 Protein claimed by the author is a combination of Tau and S100A9
It is unlikely, as in Figure 3, the S100A9 fibrils were washed from any monomers or oligomers via centrifugation and resuspension. There is a small possibility of some residual or disaggregated S100A9 in the solution, however, if it interacted with S100A9 fibrils, it would have already done so prior to the addition of Tau and we would not see any additional effects.
9. Why higher ionic strength causes the majority of aggregate clusters to dissociate? There is such a high higher ionic strength under physiological conditions?If not, what is the significance of the research?
Higher concentrations of salt ions cause electrostatic shielding between both proteins, which have opposite charges under the experimental condition pH. The key point is that “a majority” of clusters dissociate, while part of them remain stable. We have added additional information regarding the ionic strength within neurons, glial cells and in the cerebrospinal fluid. The manuscript Discussion section also emphasizes that this interaction and cluster formation can be heavily impacted by physiological condition ionic strength.
10. The result section fails to present the results in the picture very well
We have made modifications to the Figure descriptions and all parts of the text to increase its clarity. Modifications were also made based on other Reviewer and Editor comments.
11. Figure legend cannot describe and summarize pictures well
All Figure legends were modified to include all missing information (protein concentrations, number of repeats, statistical information, buffer solutions). Any additional descriptive information would overlap with the main text of the manuscript.
Round 2
Reviewer 1 Report
Comments and Suggestions for Authors
The new additions to the manuscript made a big difference. The quality of the paper had improved, and all my questions were addressed. No more comments.
Author Response
Thank you very much for your decision.
Reviewer 2 Report
Comments and Suggestions for Authors
The author did not respond well to my comments, and several important academic issues still exist in the manuscript. Therefore, I do not recommend publishing in the International Journal of Molecular Sciences. My comments are following:
- Why western blot of the samples would not provide any additional relevant information? Aren't large heterogeneous clusters formed by the aggregation of various forms of small monomers or oligomers? The fluorescence and optical density measurements, as well as FTIR spectroscopy can only reflect the final state. You need to determine that the aggregate is formed only by S100A9 rather than the tau or S100A9-tau complex.
- How does Tau enhance the aggregation of S100A9 protein? Electrostatic adsorption?
- Purity information of the protein should be provided, such as SDS-PAGE images or HPLC results
- To demonstrate that tau enhances the aggregation of S100A9 protein, I suggest that the authors canlabel tau and S100A9 to explore the mechanism by which it induces aggregation.
- If Optical density measurements are also a commonly used method to detect protein aggregation or fibril self-association into larger clusters, please cite the relevant reference.
- In the case of S100A9 control samples, all conditions resulted in a similar gradual increase of ThT fluorescence intensity (Figure 1B), with the 100 mM NaCl sample having a significantly lower end-point intensity. Without statistical analysis, I have no idea how the author reached such a conclusion. Furthermore, similar conclusions have repeatedly appeared in other parts of the manuscript, raising doubts.
- The reviewers were unable to obtain the image of “Aggregated with Tau” from Figure 1E, which shows large aggregate cluster formation from the worm-like fibril interaction with tau than “control”. Perhaps high-resolution TEM images are needed for support. However, I think the change in the optical density of the sample is not sufficient to support the formation of large particles.
- In fact, tau protein can aggregate in the presence of heparin and other inducers. The author claimed to have used 12.5 μm (over 560 μg/mL) of tau for all experiments. Will such a high concentration cause self-aggregation?The author needs to provide the corresponding results.
- Is it possible that the Tau Enhances Aggregation of S100A9 Protein claimed by the author is a combination of Tau and S100A9
- Author responsed that higher concentrations of salt ions cause electrostatic shielding between both proteins, which have opposite charges under the experimental condition pH. The key point is that “a majority” of clusters dissociate, while part of them remain stable. What is the isoelectric point of tau and S100A9? In fact, both Aβ42 and tau were induced to aggregate in PBS (0.01M) containing 137 mM NaCl. How do the authors explain this?
- More importantly, all the TEM and AFM images provided by the authors are selective and cannot confirm the authors' conclusions. When the protein concentration is 12.5 μM, the TEM images in the Figure 3E are overly biased. If large protofibril clusters were formed, these protofibrils should be insoluble. The authors could further support this conclusion by testing the soluble protein content. Moreover, the label of “F” was missing in the Figure 3.
- The results section fails to present the results in the Figures.
Author Response
The response is attached with the pdf document that includes graphical images. Thank you!

Round 3
Reviewer 2 Report
Comments and Suggestions for Authors
Thank you for replying to my comments, the question I am most concerned about is whether tau is involved in Aggregation of S100A9 like “glue” as author said. Secondly, If involved, what form is tau in S100A9 fibril clusters? As the author stated, tau itself does not aggregate, but it does aggregate in the presence of seeds. I'm not sure if S100A9 fibril is this seed.
In Comment 1 author responded that the S100A9 fibril clusters contain Tau, but not in a classical amyloid fibril type of way. Whether the Tau is monomeric or oligomeric is difficult to determine.
First of all, the author needs to use data to confirm that the S100A9 fiber cluster contains tau protein, so as to confirm the author's conclusion (Tau Enhances Aggregation of S100A9 Protein ). I think immunostaining can confirm whether tau protein is present in the S100A9 fiber cluster. In addition, fluorescent labeling can further respond to Comment 4 and the author's view that tau can be used as glue to promote the aggregation of S100A9
Then clarify whether it is a monomer or an oligomer. As the author stated, tau itself does not aggregate, but it does aggregate in the presence of seeds. I'm not sure if S100A9 fibril is this seed. The western blot can confirm it because it can distinguish between Aβ monomers and oligomers. Moreover, it can further respond Comment 9, 11